# Bio-Composites Consisting of Cellulose Nanofibers and Na^+^ Montmorillonite Clay: Morphology and Performance Property

**DOI:** 10.3390/polym12071448

**Published:** 2020-06-28

**Authors:** Runzhou Huang, Xian Zhang, Huiyuan Li, Dingguo Zhou, Qinglin Wu

**Affiliations:** 1College of Materials Science and Engineering, Nanjing Forestry University, Nanjing 210037, China; runzhouhuang@njfu.edu.cn (R.H.); shinezhang17@163.com (X.Z.); feiqia@126.com (H.L.); dgzhou@njfu.edu.cn (D.Z.); 2School of Renewable Natural Resources, Louisiana State University Agricultural Center, Baton Rouge, LA 70803, USA

**Keywords:** cellulose nanofibers (CNFs), Na^+^ montmorillonite, composite films, fire retarding, wettability, thermal decomposition kinetics

## Abstract

This paper reports the usage of cellulose nanofibers (CNFs) as a continuous nanoporous matrix and nanoclay (NC) as additive to fabricate hybrid films. CNF/Cloisite Na+ nanoclay composite films containing 10–50 wt % of NC were prepared for the study. The effects of NC incorporation and its content on mechanical, wettability and thermal degradation properties were investigated. The results showed that the film had a multilayer structure with gradually deposited CNT-NC hybrid on the filter paper Pure CNF films had higher moduli compared with those from the composite films, as the incorporation of NC decreased hydrogen bonding and networking ability of CNFs, especially at the high NC loading levels. The composite films demonstrated self-extinguishing ability when being exposed to the open flame. Composites with over 35 wt % NC did not burn because of the formation of a protective barrier containing ordered NC platelets. The addition of montmorillonite NC led to increased surface water contact angle, showing enhanced hydrophobicity of the material. During the film’s thermal pyrolysis, the first process occurred between 100 and 200 °C, resulting mainly from the evaporation of absorbed water; the second, between 280 and 350 °C, indicated thermal decomposition of cellulose; and the slow third stage happened from the 350 to 600 °C, representing carbonization. The results demonstrate that the apparent activation energies for all the CNF/NC composites were higher than the pure CNF film. CNF/NC films fabricated in this process are a promising barrier material for packaging applications.

## 1. Introduction

Combining inorganic and organic polymer components is a convenient way to fabricate composite materials, which have improved mechanical and barrier characteristics [1]. Such materials include hybrid systems containing layered clay minerals [2]. In general, nanoclays (NCs) possess a high aspect ratio as well as plate-like particle morphology, and both properties are suitable for use as efficient reinforcing fillers [3]. Hybrid materials containing organic polymers and layered NCs are not only flexible and easy to fabricate and process, but also offer fire retardancy and excellent wettability properties without compromising their mechanical integrity [4,5,6,7].

As a natural high-molecular-weight polymer, cellulose is one of the most abundant and renewable materials. It is estimated that 100 billion tons of cellulose are generated annually [8,9]. Significant research efforts have been undertaken to demonstrate that cellulosic fibers can substitute inorganic fillers to reinforce commercially available matrix materials such as plastics [7]. Natural fibers offer advantages, such as abundance, low density, biodegradability, and renewability as well as low cost. Yet, widespread industrial applications of lignocellulosic and other natural fillers are still limited, partially because of the overall poor mechanical performance of the filled composites despite the fact that mechanical properties of stand-alone cellulosic fibers are often very good [10].

Research on the utilization of nanofibers from cellulose has been widely reported [7]. A combination of cellulose nanofibers (CNFs) and NCs with a nacre structure was successfully demonstrated by [9] in a layered composite with unique properties. Cerruti et al. [3] synthesized a hybrid nanomaterial (using 3% of montmorillonite clay and 97% of regenerated cellulose) with extraordinary thermal-oxidative properties. Such material can be used for applications where fire-resistance and fire retardancy are needed. Sehaqui et al. [6] successfully prepared 50CNF/50NC film, and Liu et al. [9] fabricated CNF films containing 50–90% of NC using a process similar to the paper-making method. Materials containing nacre-like structures also demonstrated self-extinguishing performance. Chen et al. [5] modified softwood kraft fibers with alkylammonium clay and studied the moisture sensitivity of the resulting composite materials. The treated kraft fiber mat had a hydrophobic surface. Despite a variety of literature reports dedicated to CNF/NC composites, only a few studies have focused on nacre-mimicking nano-composites with lower NC contents. Thus, this paper reports the usage of CNFs as a continuous nanoporous matrix and NC as additive for fabricating and characterizing hybrid CNF/NC films with NC content less than 50 wt %. Such materials can find applications in packaging, coating, liquid filtration and purification, and other advanced electronic devices. The paper-making method was chosen to make the films because of its simplicity and versatility. We demonstrated how NC content affected fire retardancy, wettability, thermal decomposition, and mechanical properties of the resulting composite material. In addition, three kinetic methods (Kissinger [11], modified Coats–Redfern [12] and Flynn–Wall–Ozawa methods [13]) were employed to evaluate the apparent activation energies among various materials.

## 2. Experimental Procedures

### 2.1. Raw Materials and Sample Preparation

Bleached cellulose nanofibers was purchased from the Nippon Paper Chemical Co. (Tokyo, Japan). Samples of the original CNF gel (25 wt % solid) was first diluted with DI water to form 0.5 wt % suspension. The suspension was then homogenized nine times at 207 MPa pressure using a model M-110EH-30 microfluidics machine (Microfluidics Corp., Newton, MA, USA). The resulting material was CNF hydrocolloid for use in the study.

Cloisite Na + NC (with the cation-exchange capacity equal to 92 mEquiv/100g and average particle size of ~110 nm) was purchased from Southern Clay Products Inc. (Gonzales, TX, USA). The 0.5 wt % NC suspension consisting of 3.5 g of NC and 696.5 mL of DI water was made and mixed thoroughly under constant stirring.

Hydrophilic dura-pore polyvinylidene fluoride microfiltration membrane (DVPP 04700) 125 μm thick and with the average pore diameter of ~0.65 μm) was purchased from Millipore Co. (Burlington, MA, USA). The filter paper (0.2 mm thick and with a pore diameter equal to 0.6 μm) was acquired from Toyo Roshi Kaisha Co. (Chiyoda-ku, Tokyo, Japan).

The hybrid suspensions with CNF/NCmixing ratios of 0/100, 4/96, 10,90,15/85, 20/80, 35/65 and 50/50 wt % were made. The resulting amber-colored mixtures were then filtered under vacuum over the paper filter paper to remove water and form CNF/NC mats. Each formed NC-CNF CNF/NC wet mat was carefully removed from the top of the filter paper and was placed between two polished stainless steel plates. The assembly was dried under weight at 70 °C for 24 h in a vacuum oven (LC-233B1, Espec Co., Hudsonville, MI, USA). The CNF/NC films with thicknesses of about 1 mm were obtained (Figure 1).

### 2.2. Characterization of CNFs and CNF/NC Composite Films

A JEOL 100 CX machine (JEOL USA Inc., Peabody, MA, USA) was used to conduct transmission electron microscopy (TEM) to analyze the CNF morphology with an accelerating voltage of 80 kV. Photoshop image processing software (Gatan, Pleasanton, CA, USA) was utilized to calculate fiber diameters based on the obtained TEM images. A Quanta TM 3D FEG dual beam device was used for field-emission scanning electron microscopy (FE-SEM) to analyze the cross-section morphology of the film samples (FE-SEM, a FEI QuantaTM 3D FEG dual beam SEM/FIB system, Hillsboro, OR, USA). Before the SEM analysis, sample films were immersed into liquid nitrogen to render their brittleness. Later, the samples were impact broken into two halves for cross-section exposure, and the cross-section area was subsequently coated using gold for minimizing the charging of the sample.

A Tensor-27 analyzer (Bruker, Billerica, MA, USA) was utilized to conduct Fourier transform infrared (FTIR) spectroscopy. The Zn/Se ATR crystal cell was used to take each spectrum under ambient temperature at the transmittance mode. In order to collect one spectrum, typically we recorded 64 scans within the range of 4000–500 cm^−1^ under the resolution of 4 cm^−1^. Each composition was analyzed by FTIR using three different samples or parts of the sample.

Wide-angle X-ray diffraction (WXRD) patterns were recorded at room temperature with the application of a Bruker D5000 instrument (Bruker, Billerica, MA, USA). CuKα radiation was used as the X-ray source. During data collection, the instrument was operated at 40 kV and 30 mA. The XRD data were recorded for the 5–40° 2θ angle range at 0.6°/min scan speed. MDI Jade (v. 6.5.26) software (ICDD, Newtown Square, PA, USA) was used for XRD data processing.

The limiting oxygen index (LOI) of the films was determined by igniting 10 × 0.65 cm sample pieces in a glass tube containing O_2_/N_2_ mixture. The test pieces were mounted at a 45° angle and exposed to the flame for 1 s. Then, the flame was removed, and the O_2_:N_2_ ratio, where the sample continued to burn for at least 30 s, was recorded. The LOI was then calculated using the formula below:LOI = Volume of O_2_/Volume of (O_2_ + N_2_) × 100(1)

Film surface wettability was assessed with surface contact angle measurements using a liquid contact angle tester (First Ten Angstroms FTA 200, First Ten Angstroms, Inc. Portsmouth, VA, USA). For this purpose, droplets of deionized water (3–5 μL) were placed at three different locations along the longitudinal line in the middle of a sample. The data were recorded at 15, 30, 45, and 60 s and then used to analyze drop shape progression using Excel. To eliminate the decreasing rate effect, a linear extrapolation of the contact angle vs. time plot to the zero time point was applied to obtain contact angle value at the zero time point.

The sample density was calculated using sample weight and dimensions. A digital dial indicator (Mitutoyo, Tokyo, Japan) was used to measure sample film thickness, width and length, and an electronic analysis balance was used to measure weight. Porosity was calculated using the equation shown below:Porosity = [(*ρ*_t_ - *ρ*_a_)/*ρ*_a_] × 100(2)
where *ρ*_a_ and *ρ*_t_ are the real and theoretical density of the nanopaper, and *ρ*_t_ is a sample density calculated using the densities of the constituents (ρ_CNF and_
*ρ*_NC_ are equal to 1500 and 2860 kg/m^3^, respectively (Liu et al. 2013) and their corresponding weight fractions (W_CNF_, W_NC_) as weighting factors:*ρ*_t_ = (*ρ*_CNF_ W_CNF_ + *ρ*_NC_ W_NC_)(3)

The tensile properties of the films were tested using a Universal Materials Testing Machine from Instron Co. (Norwood, MA, USA) and a 100 N load cell at 4 mm/min strain rate. The sample size was 30 mm long, 0.95–1 mm thick and 3 mm wide. Prior to the testing, the samples were conditioned at room temperature for at least 48 h to equilibrate the material at the test conditions. Each film composition was tested using five different samples. The final value is reported as an average of these five measurements, with the uncertainty calculated as standard deviation.

A Q50 thermo-gravimetric analyzer (TGA; TA Instruments Inc., New Castle, DE, USA) was employed for measuring weight loss (WL) information to indicate the thermal decomposition of the material. The samples were heated at four distinct rates (5, 10, 15, and 20 °C/min) from ambient temperature (25 ± 3 °C) to 800 °C. In the TGA test, high-purity nitrogen (0.5% oxygen and 99.5% nitrogen) was used as the protective/purging gas. Before each test, the samples (8–10 mg) were dried in a vacuum oven (LC-233B1, Espec Co., Hudsonville, MI, USA) at 70 °C for 24 h. Each composite formulation was analyzed twice under the same experimental condition. The approximate overlapping of these two curves was considered a reasonable agreement.

Table 1 shows the three common methods for determining activation energy (Ea), namely, the Flynn–Wall–Ozawa, modified Coats–Redfern and Kissinger approaches. In the Flynn–Wall–Ozawa approach, –Ea/R is obtained from the log(β/T^2^) slope as a function of 1/T. In addition, the iso-conversional Coats–Redfern approach represents the integral approach, which can determine Ea/RT based on the line slope through drawing lnβ as a function of 1/T at various conversion rates. As for the Kissinger approach, ln(β/T^2^p) is drawn as a function of 1/Tp for different experiments under various heating rates, and Tp, the peak temperature, is acquired based on a Thermal Gravimetric Analysis, TG (Derivative thermogravimetry, DTG) curve.

The Universal Analysis software (TA Instruments Inc., New Castle, DE, USA) was used to automatically obtain both DTG and TG curves. The Microsoft Excel software was employed to calculate the activation energy.

## 3. Results and Discussion

### 3.1. Morphology, Structure and Composition Characterization

TEM micrographs of cellulose nanofibers clearly showed individualized fibers in the reprocessed CNF material (Figure 2a,b). The average diameter of the CNFs was 30.4 ± 3.4 nm and the CNF length were up to several microns.

The resulting composite films demonstrated a multilayer structure (Figure 2b) with the wave-like layers positioned parallel to each other. We believe that such morphology was very likely predetermined by the flocculation process, which took place during film making [4]. Substantial flocculation occurred at high fiber contents near the top surface of the filter paper. Multilayers were formed because of the electrostatic and hydrogen-bonding interactions between the layers [9]. The irregularity of the spaces between the layers can be explained by the poor bonding between the polymer chains and NC agglomeration. The NC platelets attached to CNF only at the edge of NC platelets by electrostatic interactions because NC has negatively charged surface but positively charged edges, which attract negatively charged CNFs [14].

The FTIR spectra of the CNF-NC composite materials containing 50% and 0% of NC (Figure 3d,e, respectively) showed weak bands at 3628 cm^−1^, which correspond to the interlayer hydrogen bonding in the clay confirming its hydrous nature and hydroxyl linkage presence. All CNF composites demonstrate peaks at ~3335 cm^−1^, which correspond to hydrogen-bonding in the cellulose (see Figure 3a–d). The intensity of this peak decreased as the CNF content increased because of a higher density of the cellulose hydroxyl groups and hydrogen bonds. Additionally, we believe that this indicates that the hydrogen bonding between cellulose fibers in CNF was affected by the presence of the NC platelets. The characteristic peak corresponding to the C–H stretching vibrations of methyl and methylene groups also shifted from 2904 to 2895 cm^−1^ as the amount of NC in the composite decreased. Absorbance at 895 cm^−1^, assigned to C–O–C asymmetric stretching in β-(1–4) glucoside linkages, disappeared as NC content in the composite increased. Bands at 1643 and 1631 cm^−1^ corresponded to O–H bending of adsorbed water in CNF and NC materials, respectively [15]. Weak peak at 794 cm^−1^ was attributed to the Si–O, (Al, Mg)–OH stretching of clay [16,17]. The presence of this peak might also indicate the formation of a hydrogen bond between OH group of CNF and (Al, Mg)–OH group of an NC platelet [18,19]. However, the weak form of this FTIR peak suggests that this interaction is not very strong, which agrees with our other results discussed below.

As shown in Figure 4, CNFs had a main characteristic peak angle of 22.6° (002), and two smaller peaks at 15.1° and 16.6°, respectively, representing the (001) and (110) planes of the crystalline core. Compared with nano-montmorillonite clay (NC), the first characteristic peak of the composite material became larger and shifted to a lower angle, indicating that there was a chemical bond between CNF and NC. As shown in Figure 4, the angle of the first diffraction peak (001) of the composite film was at 2θ = 6.04°, 5.95°, 5.68°, and 5.78° for 90CNF/10NC, 80CNF/20NC, 65CNF/35NC, and 50CNF/50NC materials, respectively. This is because the NC is dispersed in the water and expands, causing the nanofibers to enter the interior of the NC, thus increasing the gap degree and causing the diffraction angle to shift to a low-angle area.

The first characteristic peak of all materials was relatively narrow. The second characteristic peak of NC was at 28.48°, but the second characteristic peak of the NC material did not appear in the XRD patterns of CNF and films with a low content of NC. However, when the content of NC increased to 35% and 50%, the second characteristic peak of the material appeared and was shifted to 28.62°. The height of the overlapping peaks at 15.1° and 16.6° of CNF decreased with the increase in the NC content. Especially when the content of NC increased to 35%, the first characteristic peak disappeared, and the second characteristic peak shifted to 17.3°. The XRD diffraction angle at 22.6° is the diffraction angle of the cellulose crystals in CNFs. The characteristic peak of CNF at 34.5° disappeared due to the chemical bond between NC and CNF. This also indicates that the amorphous area of the composite material increased.

### 3.2. Flame Retardant Properties of the CNF/NC Films

Flame retardant experiments showed that pure CNF film had a similar LOI value (21.8) as that of regular printing paper (21.7) but had a higher value than that of the filter paper (19.9) (Figure 5a). All CNF/NC films fabricated in this work had significantly higher LOI values, and LOI values increased as NC content in the composite increased (Figure 5a).

Pure CNF film burned quickly and completely while samples with low NC ratio self-extinguished after the torch flame was removed. Such behavior can be explained by the fact that upon exposure to flame, the NC platelets formed a continuous protective layer, which prevented oxygen penetration and heat diffusion slowing down oxidation kinetics. It is also possible that simultaneous NC layer and char formation also contributes to the film resistance to flame [9,14]. At NC contents in the range of 35–50%, the films could not be burned even in the oxygen-rich environment, very likely because of the formation of a thick and strong protective surface layer. A burned sample consisting of pure CNF curled but the shapes of CNF/NC films remained straight, possibly due to a more favorable char formation process (see Figure 5b). Additionally, composite films had numerous bubbles on their surfaces which demonstrated the intumescent behavior of the NC in the composite film [9].

### 3.3. Wettability of the CNF/NC Composite Film

The CNF/NC composite films had larger water contact angles than those of the unmodified CNF film (Figure 6 and Figure 7), indicating increased film hydrophobicity. The contact angle of the CNF film was 55.9°, and the surface contact angle of the film after the addition of NC increased. The angles of CNF/NC films containing 5% and 10% of NC were about 58° and of those containing 15% and 20% of NC were about 65°. The enhanced wetting behavior was observed for the composite films containing 35 and 50 wt % of NC with their contact angles of 80.7° and 82.8°, respectively (see Figure 6).

During the first 60 s, a water droplet on the CNF/CN film had a circular shape (Figure 7). The water contact angle for the composite film containing 50 wt % of NC did not change significantly during the first 60 s, while the angle for the unmodified CNF film decreased by over 3%. The water contact angles for the composite films containing less than 20 wt % of NC also changed by ~3% during the first 60 s of the experiment.

Thus, the polarity of the CNF changed after the clay was added. The hydrophilic nature of the CNF films might lead to their low resistance to water. The creation of barriers and torturous clay paths could hinder water diffusion through the matrix [2,20]. This is why the incorporation of a moderate amount of NC can be beneficial for improving CNF moisture absorption properties of the films.

### 3.4. Porosity and Tensile Properties of the Composite Films

The porosity of the composites correlated with NC content (Table 2). Unmodified CNF contains disordered pores [9]. The porosity of the composite CNF/NC films increased as NC loading increased [21,22].

The addition of NC significantly affected tensile strength and elastic moduli of CNF-based films (Table 2). The lowest tensile strength (53.8 MPa) was for the CNF/NC composite containing 50% of NC. Thus, poor tensile strength was very likely because of aggregation of un-exfoliated NC as well as poor dispersion of NC in the CNF matrix [23]. The major load-carrying phase in the composite is a continuous CNF network. It is reasonable to believe that the agglomerated NCs may cause stress concentration in the composite, decreasing its total tensile strength. The results obtained by others are also in accordance with this point of view [9]. Pure CNF film also showed higher tensile modulus than that of the composite films. The incorporation of NC decreased the natural ability of CNF fibers to create ordered hydrogen bonds [23].

### 3.5. Thermal Degradation of the CNF/NC Composite Film

Figure 8 presents the DTG and TG curves for CNF as well as CNF with NC. As observed, each sample showed a low WL at low temperatures (100–210 °C), which was associated with absorbed water evaporation. The initial degradation temperature of the composite film was lower than the initial degradation temperature of the CNF film. At high temperatures (300–360 °C), CNF pyrolysis as well as char formation (>360 °C) contributed to the major degradation behaviors.

It is easily observed from Figure 8 that char residue increased with the increase in the NC level. Inorganic NC material greatly affected charring in the film. In Figure 8b, the degradation behaviors of pure CNF and CNF/NC film are similar, except the height of the degradation peak. Compared to CNF, the CNF/NC films showed a lower and smaller peak. This indicates that the degradation rate of CNF/NC film is much weaker than that of CNF. During the pyrolysis, the first process occurred between 100 and 200 °C, resulting mainly from the thermal decomposition of samples; the second between 280 and 350 °C indicated thermal cracking of cellulose; and the slow third stage happened from the 350 to 600 °C, representing carbonization (Figure 9).

The decomposition characteristic parameters of CNFs/NC are shown in Table 3. The critical temperature point is known to shift to the greater values as the heating rate increases [23,24,25]. For the sake of preventing the linear heating influence on measuring the typical temperature variables, we extrapolated four temperatures from the four heating rates to the 0 °C/min heating rate. Moreover, we acquired the WL percentage through taking the average of four related values as well as standard deviations (SD), as summarized in Table 3.

The parameter T_0_ represents an onset cellulose decomposition temperature, which is around 270 °C. The pure CNF film has the highest value for cellulose and WL in this period is around 4%. CNF/NC samples showed lower values, indicating comparatively less initial decomposition. The parameter Tp represents the maximum decomposition temperature of the main CNF, which mainly happens at 310 °C. From Table 2, the plain film has higher Tp value with respect to other composites. The WL at this point, indicated by parameter WLp, remained in the range of 14–17%. Due to possible inorganic components occurred in the surface layer, the composites experienced smaller decomposition as the NC loading increased during this period. From peak to shift temperature Ts, CNF/NC composites had a slow degradation, as shown in Figure 8b. CNF completed almost 50% WLs at 346 °C while CNF/NC composites showed less values. Residue char was obtained when the sample was further heated to 800 °C. CNF that had a greater NC loading had significantly increased residue weight as a result of the greater inorganic content. Differences in WL and temperatures between shift point and offset (namely, WLs − WLo and Ts − To), indicated that the thermal decomposition range and char level decreased with the increase in NC level, indicating the slow thermal decomposition.

As shown in Table 4, the Kissinger method led to an apparent Ea range from 150 to 225 for all films. Kissinger’s method is a special case in determining Ea, and it may not display the overall trend of Ea due to the fact that only data from a certain conversion rate is used. CNF with inorganic NC normally shows higher activation energies than control, while higher clay loading shows a higher one (Figure 10).

Such finding was also confirmed through results obtained from the modified Coats–Redfern and Flynn–Wall–Ozawa approaches, with the ranges of about 180 and 220. With regard to CNF/NC composites, our observations suggested larger activation energy value of thermal decomposition of each organic/inorganic hybrid composite compared with these of CNF film within the range reported.

Noticeably, the values obtained from Kissinger’s approach decreased on the whole compared with those obtained using the iso-conversional approaches, whereas those from iso-conversional approaches almost remained constant. Nonetheless, the diverse kinetic analysis approaches are complementary [23,26,27,28].

## 4. Conclusions

The effects of NC incorporation and its content on the mechanical, thermal, fire retardant and wettability properties of CNF-NC film were studied. The following conclusions can be reached:1)Better strength and elastic modulus were observed for films at low NC contents. Excess NC loading led to reduced strength properties due to NC aggregation in the film.2)These composite films with about 35% NC demonstrated self-extinguishing ability when exposed to the open flame. Composites with over 35 wt % NC content did not burn because of the formation of a protective barrier containing ordered NC platelets.3)The addition of montmorillonite NC increased the hydrophobicity of the material.4)During the film thermal pyrolysis, the first process occurred between 100 and 200 °C, resulting mainly from evaporation of absorbed water; the second between 280 and 350 °C indicated thermal decomposition of cellulose; and the slow third stage happened from the 350 to 600 °C, representing carbonization. The results demonstrate that the apparent activation energies for all the CNF/NC composites were higher than the pure CNF film.5)The CNF/NC films fabricated in this work seem promising as barrier materials for packaging applications.

## Figures and Tables

**Figure 1 polymers-12-01448-f001:**
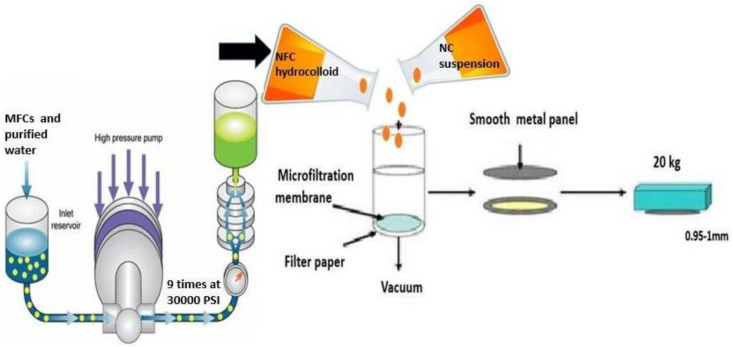
Process of making cellulose nanofiber/nanoclays (CNF/NC) composite films.

**Figure 2 polymers-12-01448-f002:**
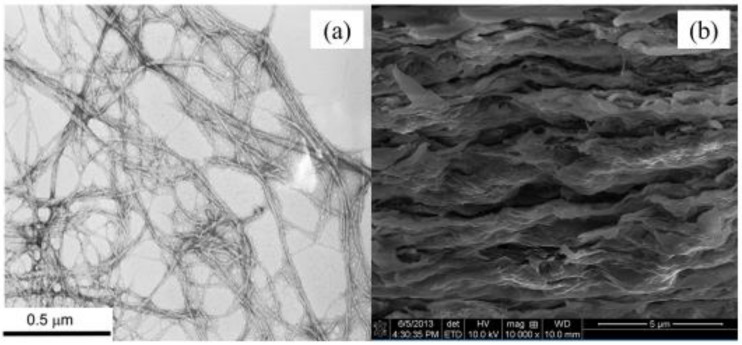
(**a**) Transmission electron microscopy (TEM) micrograph of the pure CNF and (**b**) scanning electron microscopy (SEM) micrograph of the 90CNF/10NC composite films.

**Figure 3 polymers-12-01448-f003:**
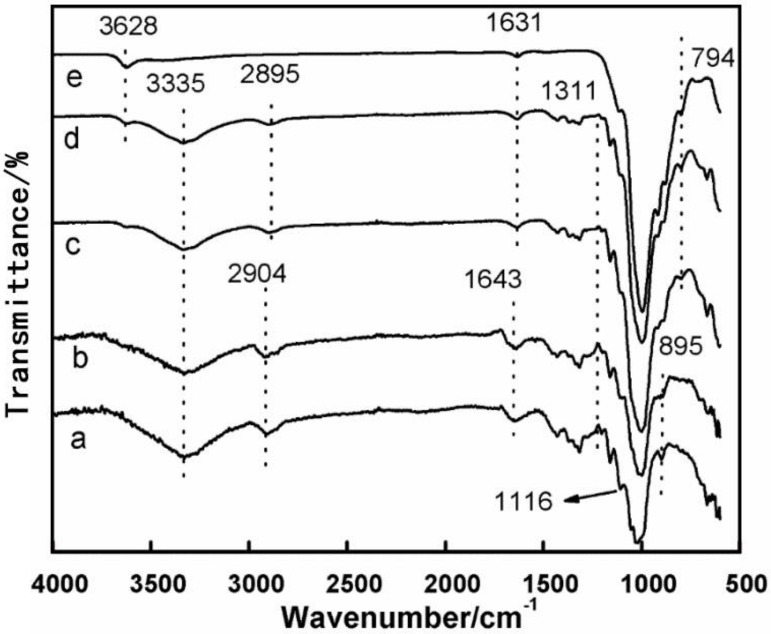
Fourier transform infrared (FTIR) spectra of CNF/clay films: (**a**) CNF film, (**b**) 80CNF/20NC, (**c**) 65CNF/35NC, (**d**) 50CNF/50NC, (**e**) NC.

**Figure 4 polymers-12-01448-f004:**
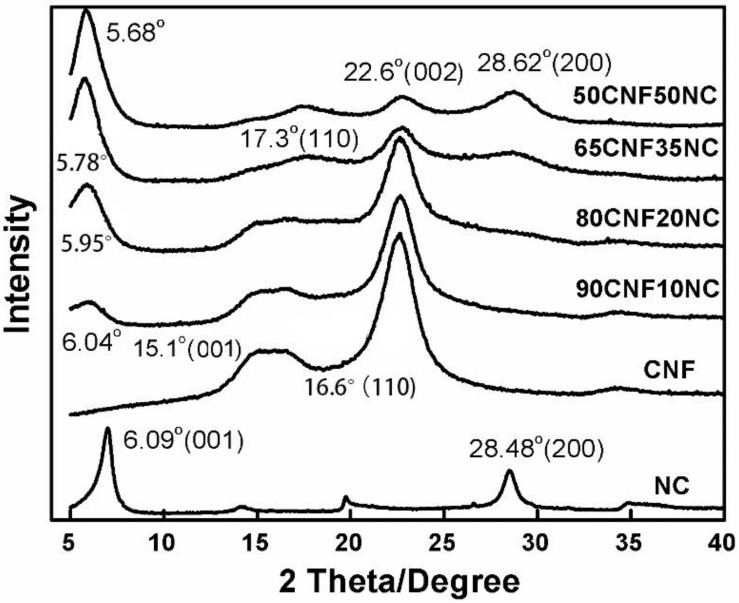
X-ray diffraction (XRD) patterns of pure NC and CNF as well as of the CNF/NC composites.

**Figure 5 polymers-12-01448-f005:**
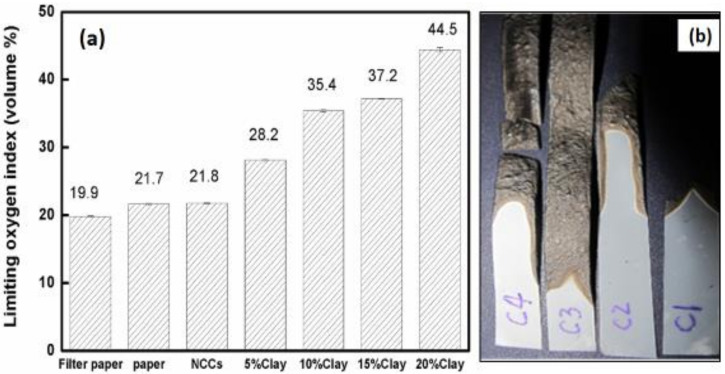
(**a**) Limiting oxygen index (LOI) of CNF/NC composites in comparison to printing and filter paper. (**b**) Photograph showing char formation for the composites containing (C1) 5%, (C2) 10%, (C3) 15%, and (C4) 20% of NC.

**Figure 6 polymers-12-01448-f006:**
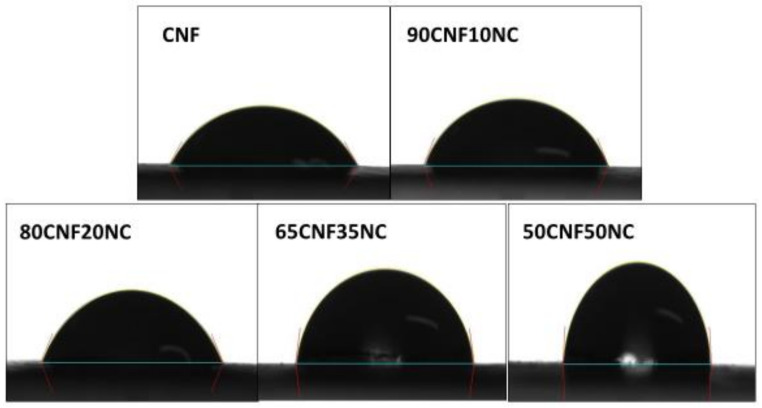
Surface water contact angle of CNF-NC films containing 0, 10, 20, 35 and 65 wt % of NC.

**Figure 7 polymers-12-01448-f007:**
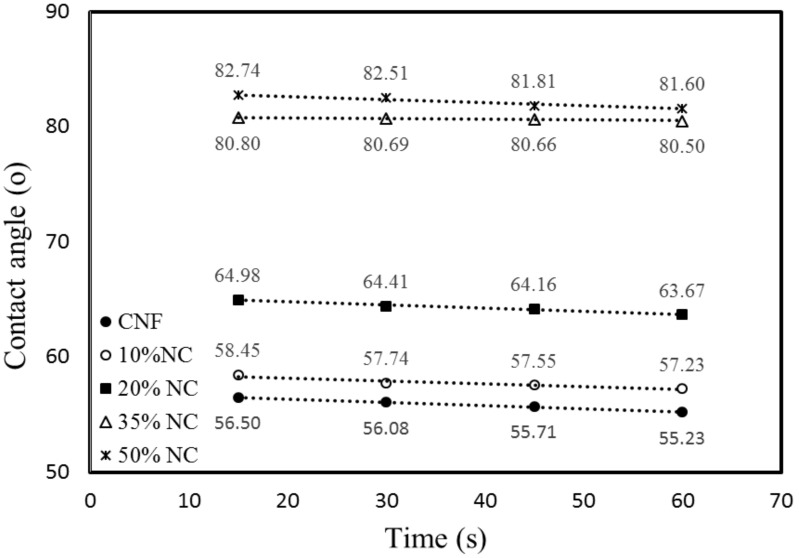
Evolution of the water contact angle of CNF-NC films containing different amounts of NC as a function of time.

**Figure 8 polymers-12-01448-f008:**
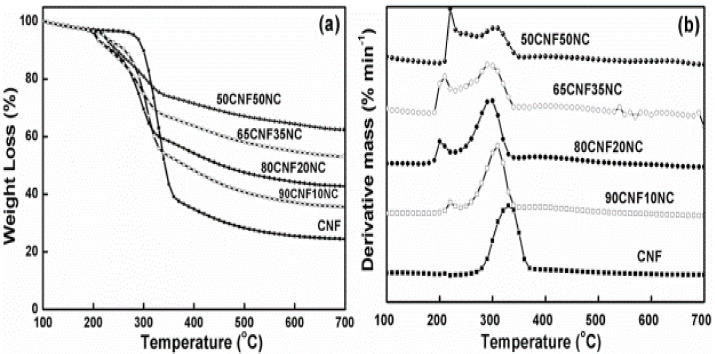
TG (**a**) and DTG (**b**) curves of CNFs/NC composites at a heating rate of 10 °C/min.

**Figure 9 polymers-12-01448-f009:**
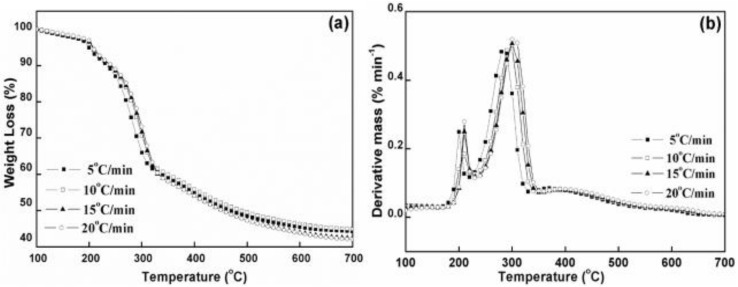
TG (**a**) and DTG (**b**) curves of CNF with 20 wt % clay under an air flow rate of 5, 10, 15 and 20 °C/min.

**Figure 10 polymers-12-01448-f010:**
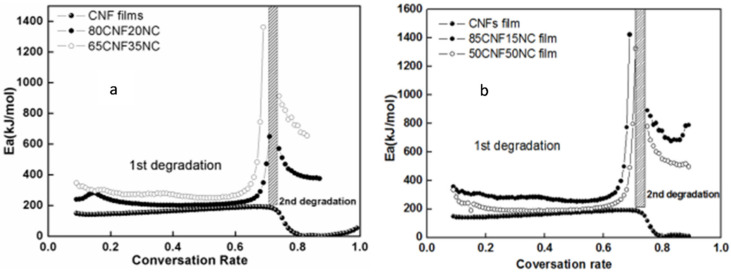
Ea for selected samples calculated by (**a**) modified the Coats–Redfern (**b**) Flynn–Wall–Ozawa method.

**Table 1 polymers-12-01448-t001:** Methods for determining Ea of CNF/NC composites *.

Method	Expression	Plots	Ref.
Kissinger	ln(β/T^2^p) = ln(AR/Ea) + (1/T)(−Ea/R)	ln(β/T^2^p) against 1/Tp	[11]
Coats–Redfern (modified)	log β = log (A Ea/Rg(α)) − 2.315 − 0.4567Ea/RT	logβ against 1/T	[12]
Flynn–Wall–Ozawa	ln(β/(T^2^(1 − 2RT/Ea ln(1 − α))) − Ea/RT	ln(β/T^2^) against 1/T	[13]

* α is the weight conversion degree, β is the heating rate, R is the general gas constant (8.314 J ⁄K∙mol), T is the temperature, and Tp is peak temperature.

**Table 2 polymers-12-01448-t002:** Density, porosity, tensile strength, modulus and water contact angles of pure and NC-modified CNF films.

NC Content (wt%)	Density (kg/m^3^)	Porosity (%)	Tensile Strength (MPa)	Tensile Modulus (GPa)
0	1305.04	13.00	121.0 ± 7.8	12.39 ± 0.83
10	1366.13	16.50	97.24 ± 3.85	9.30 ± 1.32
20	1386.56	21.75	69.48 ± 5.31	8.80 ± 1.15
35	1496.35	24.27	61.09 ± 5.97	6.02 ± 0.74
50	1589.21	27.10	53.77 ± 3.68	4.63 ± 0.83

**Table 3 polymers-12-01448-t003:** Decomposition characteristic parameters of CNF/NC composites.

Sample	T_01_ ^a^ (°C)	WL_01_ (%)	T_02_ (°C)	WL_02_ (%)	T_p_ (°C)	WL_p_ (%)	T_s_ (°C)	WL_s_ (%)	T_s_ − T_02_ (°C)	WL_s_ − WL_02_ (%)	Residue (%)
90CNF10NC	208	3.7	240	7.4 (0.4)	288	20.2 (3.5)	318	39.2 (2.0)	78	31.9	36.2
80CNF20NC	197	3.3	231	9.1 (0.3)	281	19.6 (2.5)	316	36.7 (1.1)	85	27.6	42.9
65CNF35NC	198	3.2	211	6.4 (0.6)	281	19.6 (1.2)	330	31.5 (0.1)	119	25.1	53.1
50CNF50NC	197	3.1	210	5.7 (0.2)	278	19.3 (0.8)	336	28.2 (0.1)	126	22.5	62.2

^a^ T = temperature; WL =weight loss; subscript: 0 = onset; P = DTG peak; s = shift. Values from four heating rates with mean value and standard deviation. Values from five samples with mean value and standard deviation.

**Table 4 polymers-12-01448-t004:** Apparent Ea of CNF/NC composites calculated by three model-free methods.

Sample	Kissinger		Coats–Redfern		Flynn–Wall–Ozawa	
	Ea	R^2^	Ea	R^2^	Ea	R^2^
CNF	157.6	0.9862	181.6 (8.5)	0.9904	182.2 (8.3)	0.9914
90CNF/10NC	186.6	0.9996	213.2 (13.0)	0.9993	211.7 (12.6)	0.9994
80CNF/20NC	185.2	0.9881	215.6 (49.4)	0.9934	213.8 (16.8)	0.9940
65CNF/35NC	206.1	0.9957	285.2 (48.4)	0.9863	279.8 (12.8)	0.9863
50CNF/50NC	232.7	0.9352	284.5 (45.6)	0.9849	289.7 (11.8)	0.9836

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
