# Peer review of "Bio-Composites Consisting of Cellulose Nanofibers and Na+ Montmorillonite Clay: Morphology and Performance Property"

_polymers, 2020, doi:10.3390/polym12071448_

Round 1
Reviewer 1 Report
In this manuscript the authors analyzed the influence of the inorganic filler (NC) content on the fire-retardancy, wettability and mechanical properties of the resulting composite material with nanofibrillated cellulose. Different kinetic methods/models were employed to evaluate the apparent activation energies and results were discussed. The manuscript is well written, in a clear and organized way. It contains new and interesting results, and deserves publication. I only have some minor suggestions which are highlighted in yellow in the attached pdf file. One additional comment is detailed below. I recommend the publication, the additional review is not necessary.
It is claimed that the irregularity of the spaces between the layers shown by the authors can be explained by the poor bonding between the polymer chains and NC agglomeration. As mentioned in the paper, the size of one NC particle is about 110nm. If the agglomeration takes place, the size of these agglomerates should be even larger, and, in any case, much larger than the individual fiber diameter. Does the NC agglomeration influence the final composites properties, is it possible to study it, or discuss at least? The authors mentioned that the “Agglomerated NCs create stress concentrations in the polymer matrix, decreasing its total tensile strength”. I think it would be extremely interesting to further investigate the aggregation effects on the final composite mechanics.

Author Response
Responds to reviewer 1: 1. The reviewers comment: In this manuscript the authors analyzed the influence of the inorganic filler (NC) content on the fire-retardancy, wettability and mechanical properties of the resulting composite material with nanofibrillated cellulose. Different kinetic methods/models were employed to evaluate the apparent activation energies and results were discussed. The manuscript is well written, in a clear and organized way. It contains new and interesting results, and deserves publication. I only have some minor suggestions which are highlighted in yellow in the attached pdf file. One additional comment is detailed below. I recommend the publication, the additional review is not necessary. It is claimed that the irregularity of the spaces between the layers shown by the authors can be explained by the poor bonding between the polymer chains and NC agglomeration. As mentioned in the paper, the size of one NC particle is about 110nm. If the agglomeration takes place, the size of these agglomerates should be even larger, and, in any case, much larger than the individual fiber diameter. Does the NC agglomeration influence the final composites properties, is it possible to study it, or discuss at least? The authors mentioned that the “Agglomerated NCs create stress concentrations in the polymer matrix, decreasing its total tensile strength”. I think it would be extremely interesting to further investigate the aggregation effects on the final composite mechanics. Respond: The effects of the agglomerated NCs is in deed a very interesting issue, as pointed out by the reviewer. However, it is currently not possible for us to carry out a in-depth research on this issue because of limitations of funding and time. We are to investigate this issue in the future. In the previous manuscript, we mistakenly stated that “Agglomerated NCs create stress concentrations in the polymer matrix, decreasing its total tensile strength”, which may be ambiguous to the readers. Actually, this statement is not one of our research results or conclusions, rather, it is a conjecture, which is to be authenticated in the future. Therefore, we have replaced this statement with “It is reasonable to believe that the agglomerated NCs may create stress concentrations in the polymer matrix, decreasing its total tensile strength. The results of the research made by Liu etc. is also in accordance with this point of view [20]. And the effect of the agglomeration shall be studied in detail in the future.” This point of view is typically supported the research made by Liu, A.D.,(Ref. [20]) , therefore, the citation to this reference is kept in the manuscript as it was. And the reviewer is also thanked for point out the format and spelling mistakes, which have all been corrected. Nanofibrillated cellulose (CNF)and Lignocellulose Nano Fibers(CNF) have very similar meaning and can be replaced by each other in some circumstances. For consistency and clarity, we use “Lignocellulose Nano Fibers(CNF)” in the whole passage of the revised manuscript. All other errors highlighted in the PDF file have been corrected.

Reviewer 2 Report
See attached file

Author Response
Dear reviewer:
We sincerely appreciate and highly value your comments and suggestions, and according to which, we have humbly paid much effort in improving the research and manuscript. We hope the revised manuscript could solve all the problems you raised and win your approva.

Reviewer 3 Report
Just a several comments, I would like to hear Authors response (these are also included in manuscript):
- Did you measured the fibers as they really are of nano scale, or just assumed that after mentioned treatment these should be nano?
- How did you prepared the samples for your tests? Did you conditioned them? Under what conditions? You provided the information about conditioning for tensile tests only.
If the samples were not oven dry, the moisture content can influence your results, not only LOI, but also wettability, flame retardancy etc. The addition of NC to NFC can result in different equilibrium moisture content of the produced composites, since the reaction to humidity by NC and NFC can vary.
Please explain the influence of samples moisture content on your results.
- What tools did you used to measure weight and dimensions (density)?
What was the precision of these?
- In tensile tests you used 90-100 micrometers thick samples. Where did you get the samples of that thickness, since you produced a samples with thickness of 0.95-1 milimeters?
- How can you explain the density change (about 22% referred to NC content 0) (table 2.)?
In every tested variant you added the same dry weight of ingredients, and the size of produced papers was the same, so density should be the same, as well.

Author Response
Dear reviewer,
We sincerely appreciate and highly value your comments and suggestions, and according to which, we have humbly paid much effort in improving the research and manuscript. We hope the revised manuscript could solve all the problems you raised and win your approval: .

Round 2
Reviewer 2 Report
The submission number 807493 of title Bio-Composites Consisting of Lignocellulose Nano Fibers
and Na Montmorillonite: Morphology, Structure, Composition Characterization, Mechanical, Fire
Retarding Wettability Properties and Thermal Decomposition kinetics, received on May 12th and
reviewed on May 18th 2020, came back to me on June 3th for a second round review, it is an honor
for me to do it.
Eventhough English style correction have been done, some other typing mistakes have emerged
blocking clarity of data analysis.
I strongly suggest the authors to read the review of Moon et al, Chem. Soc. Rev., 2018, 47, 2609 (DOI: 10.1039/c6cs00895j) in order to get clarity on concepts of lignocellulose and cellulose, even after homogenizer treatment. Nanofibrillated cellulose and Lignocellulose Nano Fibers have no very similar meaning and can not be replaced by each other under any circumstances.
I am deeply sorry not to accept this paper, again, since major misconceptions and typing mistakes
make this submission still unclear.
Author Response
We sincerely appreciate and highly value your comments and suggestions. We have made an extensive editing of the paper based on the review. We believe that the paper is more readable now.
The material that we were using was bleached cellulose nanofibers. Thus there was no lignin in it and lignocellulose term would not apply. The original material from Japan had a network structure which is often called nanofibrillated cellulose. After we reprocessed the material with a high pressure homogenization machine, we had more individualized fibers (see the TEM picture), which are called cellulose nanofibers. We have read the review by Moon et al, Chem. Soc. Rev., 2018, 47, 2609 (DOI: 10.1039/c6cs00895j) and get clarity on concepts of lignocellulose and cellulose. Currently, there is still a debate about the terminology for cellulose nanomaterial among micro/nano-fiberillated cellulose and cellulose nanofibers (CNFs). In Japan, all these materials are called cellulose nanofibers. Most literature uses CNFs for more individualized cellulose nanofibers. Since we had more individualized fibers, we chose to use CNFs in our paper.

Reviewer 3 Report
Thank you for collaboration over your manuscript and for your comprehensive response to review.
Regards!
Author Response
We sincerely appreciate and highly value your comments.
